# Amazon Deforestation and Global Meat Consumption Trends: An Assessment of Land Use Change and Market Data from Rondônia That Shows Why We Should Consider Changing Our Diets

**Veronica Garcia Donoso** [1,2,*] **, Mayumi C. M. Hirye** [3,4] **, Christiane Gerwenat** [5] **and Christa Reicher** [5]

1    Architecture and Urbanism Course, Federal University of Santa Maria (UFSM),
     Cachoeira do Sul 96503-205, Brazil
2    Georg Forster Research Fellowship, Alexander von Humboldt Foundation, 53173 Bonn, Germany
3    Institute of the Environment and Sustainability, University of California, Los Angeles, CA 90095, USA;
     mhirye@ucla.edu
4    Quapá Lab, Faculty of Architecture and Urbanism, University of São Paulo, São Paulo 05508-900, Brazil
5    Institute for Urban Design and European Urbanism, RWTH Aachen University, 52062 Aachen, Germany;
     christiane.gerwenat@rwth-aachen.de (C.G.); reicher@staedtebau.rwth-aachen.de (C.R.)
*    Correspondence: veronica.donoso@ufsm.br

**Abstract:** This paper seeks to elucidate the interrelationship between global meat consumption and deforestation in the Amazon region. To this end, empirical research is conducted to investigate land use changes in Rondônia and the expansion of pasture areas and beef production. Brazil is one of the largest beef producers in the world, with products destined for local and global markets. Based on bibliographical research, the paper analyzes maps of land use change between 1985 and 2021, using Landsat satellite imagery and the MapBiomas methodology for Landsat mosaic and classification. The research shows that beef from Rondônia is primarily purchased and consumed in Brazil, but it is also bought and sold on the international market. Landsat imagery analysis shows the predominance of forest conversion to pasture in Rondônia. The results show that deforestation in the Amazon is directly linked to the growth of cattle ranching. Land use change from native forest to pasture for beef cattle production is a reality in the Brazilian Amazon, especially in Rondônia. It suggests the urgent need for more conscious consumption and production practices as well as ethical and sustainable eating habits.

**Keywords:** Amazon; land use change; pasture; deforestation

## 1. Introduction

The world's increasing population and its escalating impact on the biophysical environment raise many concerns, particularly about our eating habits. The global population is expected to reach 9.8 billion by 2050 according to the United Nations (2017) [1], with meat demand expected to surge by 73% from 2010 levels [2]. This growth places immense pressure on natural resources.

Food production and consumption significantly contribute to environmental problems, with meat production and consumption incurring some of the highest environmental costs [3,4]. Our current unsustainable food system cannot support the burgeoning population, consumption rates and associated environmental impacts [5–8]. There is an urgent need for solutions, including dietary changes.

Livestock farming uses 80% of the world's agricultural land, which includes both grazing pastures and croplands for feed production. Although deforestation is not a necessity for meat production, the two are closely linked. Livestock production contributes to deforestation globally, especially in the Global South, including Latin America and sub-Saharan

Africa, where the majority of livestock-related deforestation occurs [9]. Additionally, the rising global demand for beef is expected to increase pasture expansion, further impacting the rate of deforestation [9].

The deforestation of the Amazon stands out as an especially urgent problem. The Amazon rainforest, covering approximately 844 million hectares across nine South American countries, with almost 62% (522 million hectares) in Brazil [10], is a critical area of concern. The Brazilian Amazon, spanning nine states in northern Brazil (Acre, Amapá, Amazonas, Maranhão, Mato Grosso, Pará, Rondônia, Roraima and Tocantins), represents 49.50% of Brazil's territory and two-thirds of its forest area. Data from Mapbiomas (2021) [11] show that 82% of this biome retains its native vegetation, with 78.4% being forest.

Over the past 40 years, the Amazon has undergone significant deforestation. From 1985 to 2022, about 80 million hectares of forest were lost, an area over twice the size of Germany, with agriculture and pasture expanding by 84 million hectares [10]. Brazil accounted for 75% of this loss, with states like Pará and Rondônia leading in the forest-to-agriculture or pasture conversion [12–14].

The aim of this research paper is to illustrate the interconnection between global meat consumption and deforestation in the Amazon region. Consequently, the paper proposes that a more conscious approach to consumption on a global scale should be adopted, along with the implementation of ethical and sustainable food production practices.

The article presents findings about the growth of cattle ranching in the Amazon region, which is a consequence of the conversion of the rainforest into pasture. A case study is presented for the Brazilian state of Rondônia. The hypothesis of the research is that the observed increase in cattle ranching is the result of the increasing global demand for meat. Furthermore, the meat produced in this Amazon region is linked to deforestation and consumed worldwide. Consequently, the responsibility for deforestation is a global one. In order to test this hypothesis, land use maps and indices of meat production and export were studied and cross-analyzed, with data from 1985 to 2022.

This article is structured in the following order: an overview of deforestation in the Brazilian Amazon; an introduction to the state of Rondônia, its history and land use changes; information on cattle ranching in Brazil and Rondônia; and a discussion of the impacts of cattle ranching on the Amazon and global meat consumption.

## 2. Materials and Methods

MapBiomas land use land cover (LULC) annual maps of Brazilian territory are produced collaboratively and at low cost. They are freely available at the project's platform [15]. Collection 1 was published in 2016, and since then, each year, the MapBiomas initiative has released a new collection with improved methodology and extended time series. Current Collection 8.0 presents maps from 1985 to 2022.

The MapBiomas classification scheme is a hierarchical system with four levels of LULC classes. At Level 1, there are six classes: forest, non-forest formation, farming, non-vegetated area, water and not-observed. Level 2 comprises five classes of forest (forest formation, savanna formation, mangrove, floodable forest and wooded sandbank vegetation); five classes of non-forest formation (wetland, grassland, hypersaline tidal flat, rocky outcrop, herbaceous sandbank vegetation and other non-forest formations); four classes of farming (pasture, agriculture, forest plantation and mosaic); four classes of non-vegetated area (beach, dune and sand, urban area, mining and other non-vegetated areas); and two classes of water (aquaculture and river, lake and ocean). Agriculture is detailed in Level 3, differentiating temporary from perennial crops, and in Level 4, by product (soybean, sugarcane, rice, cotton or other temporary crop, and coffee, citrus and palm oil, as perennial crops).

The production of the data is divided among various teams, each of which is responsible for a Brazilian biome (Amazon, Atlantic Forest, Caatinga, Cerrado, Pampa, and Pantanal) or cross-cutting theme (Pasture, Agriculture, Forest Plantation, Coastal Zone, Mining, and Urban Area). The general methodology can be described as involving the fol-

lowing steps: (1) process Landsat imagery to produce annual mosaics; (2) classify mosaics using yearly training samples and a random forest classifier or U-Net convolutional neural network classifier; (3) apply spatial and temporal filters; and (4) integrate each biome and cross-cutting theme in a hierarchical order using prevalence rules, with definitions based on expert knowledge. This procedure is described in the MapBiomas General Handbook [16] and in the Algorithm Theoretical Basis Documents (ATBDs) of each biome and cross-cutting theme [17]. All processing steps are carried out in Google Earth Engine [18], and codes are available at the public GitHub repository [19].

An analysis of MapBiomas Collection 8.0 data was conducted using ~75,000 validation samples per year. Then, each sample was labelled according to MapBiomas LULC classes by experts after the visual interpretation of Landsat data, MODIS-NDVI time series, and high-resolution imagery from Google Earth (when available). Following Pontius and Millones [20], MapBiomas calculates different metrics to quantify accuracy: (i) Global Accuracy, which measures the estimate of the overall hit rate, and its complements, which disaggregates errors as (ii) Area or Quantity Disagreement and (iii) Allocation Disagreement. The global accuracy for legend Level 1 is 90%, with an annual variation of 88.3% to 90.6%. For Level 2, it is 85.8% (variation of 83.7% to 86.7%). For Level 3, the global accuracy and its annual variation resemble those of Level 2. Area disagreement and allocation disagreement are 9% and 1% in Level 1 and 9% and 5% in Level 2 and Level 3, respectively. These metrics are complemented with producer and user errors, calculated for each year and each class, with complete accuracy statistics available [21].

Data on meat products from the State of Rondônia for export were organized by country and by monthly and annual totals, from 1997 to 2022. The information was requested and generated from the Brazilian Agribusiness Foreign Trade Statistics (Estatísticas de Comércio Exterior do Agronegócio Brasileiro) and from the Brazilian Ministry of Agriculture and Livestock (AGROSTAT/MAPA) [22]. The information was collected both through the AGROSTAT/MAPA online and through the Access to Information Law directly with the agency.

Data on meat products from the State of Rondônia sold in Brazil were obtained both through online platforms and through a request to the Ministry of Agriculture and Livestock (SIPOA-DIPOA-SDA) through the Access to Information Law due to instabilities in online platforms. Data on meat products from the State of Rondônia are available on two platforms: PGA-SIGSIF, which organizes data from February 2021 to August 2023 [23] and the SIGSIF platform, which organizes data from January 1997 to January 2021 [24].

Population and livestock data were collected by the Brazilian Institute of Geography and Statistics (IBGE) and are available on the Agriculture and Livestock Portal [25] and the Cities Portal [26].

## 3. Results

### 3.1. Deforestation in the Brazilian Amazon

The Amazon is the largest tropical forest biome in the world and an ecosystem of global importance that has been under intense pressure from human activity for many decades. In 2021, forest formations (ombrophilous forests, seasonal forests and wooded savannas) comprised the majority of the Amazon biome (520 million hectares). Other natural formations (mainly open savannas, mangroves, floodable forests, wetlands and grasslands) included comprise 12% of the biome. Of the total biome territory, Brazil's forest coverage, comprising 310 million hectares, accounts for a substantial 45%, underscoring its significant contribution to the Amazon ecosystem.

The Brazilian Amazon is also home to the largest concentration of isolated Indigenous peoples in the world [27]. The unsustainable exploitation of the Amazon region and the rainforest affects the territories and lives of Indigenous peoples and illegal activities increase their vulnerability.

The Amazon region faces several challenges, including escalating deforestation driven by uncontrolled growth of agriculture and cattle ranching, along with illegal activities

(e.g., mining and forestry) related to the exploitation and extraction of natural resources. Tackling these challenges requires the formulation and implementation of public policies and international cooperation to build a consensus on the sustainable management of native forests.

From 1985 to 2022, the Brazilian Amazon lost approximately 14% of its forest area (53 million hectares. Most of this deforestation can be linked to cattle ranching (87%), followed by small percentages of conversion to agriculture, illegal mining and urban expansion. Prior to 2007, deforestation rates and the expansion of pasture lands exhibited elevated levels, gradually stabilizing thereafter (Figure 1).

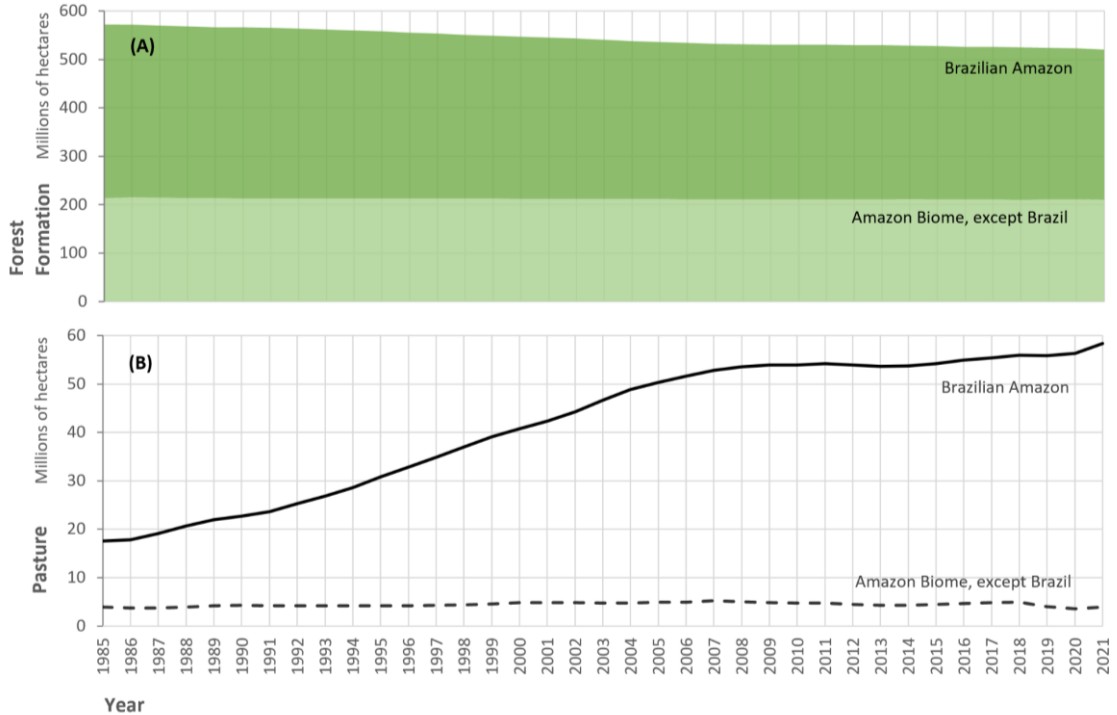

**Figure 1.** Area of (**A**) forest formation and (**B**) pasture in the Amazon biome from 1985 to 2021.

Rondônia has a surface area of 24 million hectares. It is not the Amazon's largest state, but it exemplifies the dramatic process of forest conversion (Figure 2).

Considering the land-use change analysis, the results showed that Rondônia lost 7.5 million hectares of forest in almost 40 years, between 1985 and 2022 [13], with 2003 being responsible for the most deforestation. Rondônia was responsible for almost 7% of the country's deforestation in 2022 or 139,531 hectares [11], with the capital Porto Velho being the region where most deforestation occurred in that year. The loss of forest is related to the increase in pasture and cattle herds. In 1985, forest covered 86% of Rondônia's territory. In 2021, this percentage fell to 56%, while pastureland increased from 7% of the total area of Rondônia to 36% in 2021.

Agricultural production increased in the state beginning in the 1990s, with soya plantations comprising 1.35% of the state's area by 2021. The expansion of soya farming is concentrated in the southern part of the state. In 1985, urban areas represented only 0.08% of Rondônia and by 2021 they had increased to 0.21%.

Figure 2 also shows the expansion of pastures. In the map of 1985, the great expansion of cattle ranches in Rondônia can be seen, mainly along the BR-364 highway and the roads created by this axis, entering the forest areas in a fishbone pattern of growth and deforestation.

It is important to highlight the role of Indigenous territories and protected areas in forest protection: the large remaining forest areas visible in the 2021 map in Figure 2B are within these areas.

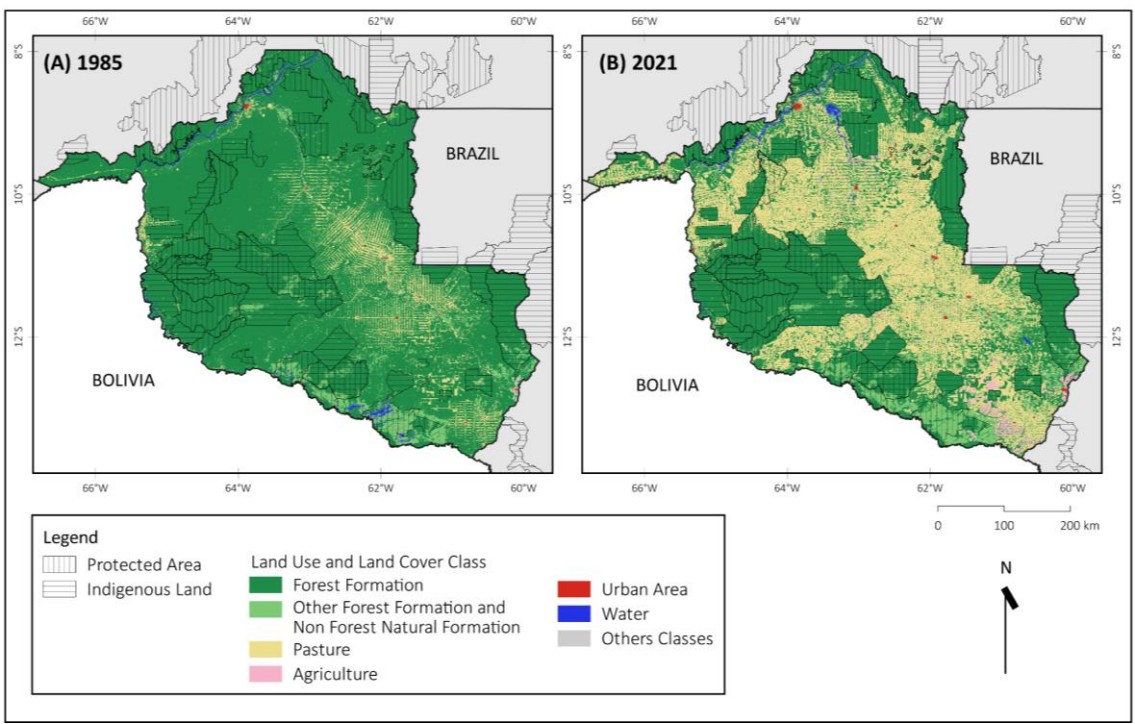

**Figure 2.** Land use and land cover in Rondônia state in (**A**) 1985 and (**B**) 2021.

The annual conversion of forest to pasture from 1985 to 2021 is shown in Figure 3. From the graph, it can be seen that the conversion of forest to pastureland peaked between 2002 and 2003, with 459 thousand hectares of pastureland. The lowest conversion period was from 2008 to 2010, with 62 and 55 thousand hectares, respectively. Between 2020 and 2021, 206 thousand hectares of forest was converted to pasture.

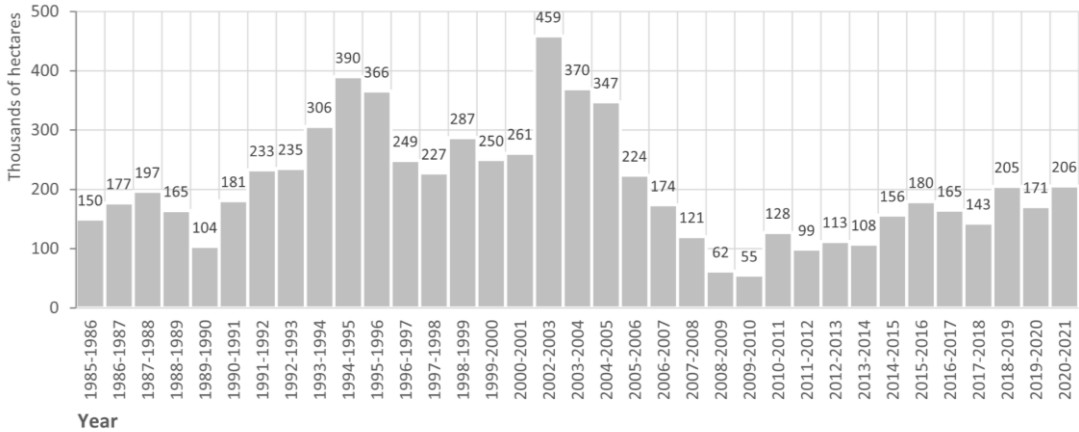

**Figure 3.** Yearly conversion of forest formation to pasture from 1985 to 2021.

As shown in Figures 2 and 3, deforestation in Rondônia is a recent process, and deforestation rates continue to increase. Thanks to the existence of protected areas and Indigenous territories, deforestation has been more stable since 2018. However, protected areas are not without pressure and many local conflicts involve illegal activities such as slash-and-burn and logging, the first step of deforestation that precedes the introduction of cattle to mark territories.

Although recent, these land-use conflicts in the state of Rondônia are not new. The misperception of the value of the rainforest and the creation of a culture of exploitation have existed in Brazil since colonial times, as discussed in the following section.

*3.2. Rondônia—History, Deforestation and Land-Use Change*

The first records of human occupation of the Amazon region show the presence of settlements approximately 14,000 years ago. They were most likely immigrants from Asia whose arrival on the American continent is explained by a number of hypotheses, including their arrival via the Bering Strait when the sea level was lower or by the navigations of Chinese admirals on expeditions around the world [28].

In the 16th century, Europeans arrived in the Amazon region and found a forest inhabited by many culturally diverse Indigenous peoples. Archaeologist Eduardo Neves estimates that there were approximately 5 million Indigenous inhabitants of the Amazon before Europeans arrived. During the European conquest of the territory, the Indigenous population was drastically reduced, mainly due to the encounter with diseases unknown to them and the use of firearms [28].

Subsequent years saw waves of migration and the occupation of the territory by settlers. In Rondônia, the construction of the Madeira-Mamoré railroad between 1907 and 1912 and the establishment of telegraph lines between 1907 and 1915 were pivotal in connecting the region and facilitating the rubber trade. These developments triggered a significant influx of immigrants from many parts of the world to Rondônia, particularly for the construction of the railway [29].

Beginning in the 1960s, the settler occupation of the state of Rondônia took on new proportions, as did the deforestation of the area for agricultural expansion. The Brazilian military government (1964–1985) encouraged the expansion of the agricultural frontier and the occupation of areas in northern Brazil, including in Rondônia [30–33].

From then on, tax incentives and investments in colonization projects and agrarian settlements accelerated the flow of migrants to the Amazon territory through the sale and distribution of land, the opening and modernization of highways and other investments in infrastructure, like the building of railroads and hydroelectric plants.

Agricultural colonization in Rondônia took place with the clearing of extensive areas of forest. This deforestation was the settlers' way of "guaranteeing" possession of the land, resulting in various conflicts and the genocide of Indigenous peoples. This migration and colonization led to a population increase of almost 1000% between 1970 and 1991. In 2021, the population was estimated at 1,815,279 inhabitants.

*3.3. Cattle Ranching in Brazil and Its Expansion in the Amazon Region of Rondônia*

The first cattle herds were introduced to Brazil by the Portuguese in 1533 in the coastal area of the state of São Paulo [34]. During the colonial period, all Brazilian territory was intensively explored, and the Amazon region was no different.

The impacts were multiple. Socially, the Indigenous and Black populations were enslaved for the exploitation of resources and affected by the introduction of diseases. Environmentally, the exploitation of natural resources led to the near extinction of valuable tree species such as mahogany and pau-brasil. Culturally, a colonial mindset prioritized resource exploitation and undervalued the forest. This mentality prevailed for many centuries and is still present today in some population groups, including in the state of Rondônia.

Extensive cattle farming predominates in Brazil. This pasture-based system has established the country as the second-largest international beef producer. In addition, Brazil's beef cattle production capacity is expected to increase until 2029 [35,36].

Brazilian meat products have been an important export product for many years. The data gathered by this research shows that, in 1997, it represented 6.82% of the country's exports, a number that increased year after year. In 2003, it represented 13.67% of the total exports, and in 2008, it reached its highest point at 20.19%. In 2022, it represented 16.16%, the second most exported product after soy, at 38.28%.

Brazil has the second-largest cattle herd in the world, with a national total of 224.6 million head in 2021. A large proportion of this cattle herd is located in the Amazon region, especially in the states of Pará and Rondônia.

The city with the largest number of cattle is São Félix do Xingu, in the state of Pará, with 2469 million head. Porto Velho, capital of the State of Rondônia, is in fourth place, with 1354 million cattle, representing 9.83% of the state's total, in the largest occupied area (3,410,000 hectares) [37].

Although the state of Rondônia only covers 2.79% of the country, it has 14.3 million head of cattle, 6.4% of the national total [38]. In 2022, Rondônia exports represented 1.39% of the total national market, compared to only 0.17% in 1998. This increase is directly related to cattle farming expansion [22].

Of the meat produced in Brazil in 2021, 25.51% was exported and 74.49% consumed domestically [39,40]. In fact, Brazil is one of the countries with the highest beef consumption in the world, at 36.4 kg/person/year, compared to 12.3 kg/person/year in Spain, the third-largest beef producer in the European Union [22,41].

*3.4. International Market: Where in the World Did Rondônia's Meat Products Go?*

Although Brazilian meat products are mostly consumed domestically, Brazil is the fifth-largest beef exporter in the world, with the main consumer market in 2021 being China (which exported 39.20% of the volume of meat produced), followed by Hong Kong (11.91%), the USA (7.52%), Chile (5.99%) and the European Union (4.19%) [42].

Rondônia's meat exports follow the same markets: in 2021, the largest international market was Hong Kong (18.93% of exportation), followed by China (18.3%) and Chile (14.21%). The United States is ranked sixth, importing 6.24% of Rondônia's meat products.

The monthly agricultural export data for the state of Rondônia between 1997 and 2023 (until July) are summarized in a diagram (Figure 4). The diagram visualizes the three largest importers of meat products per year in kilograms, and all other export countries of the year are shown as "others". It is particularly striking that Hong Kong appears 22 times, Egypt 16 times and Russia 7 times in these statistics.

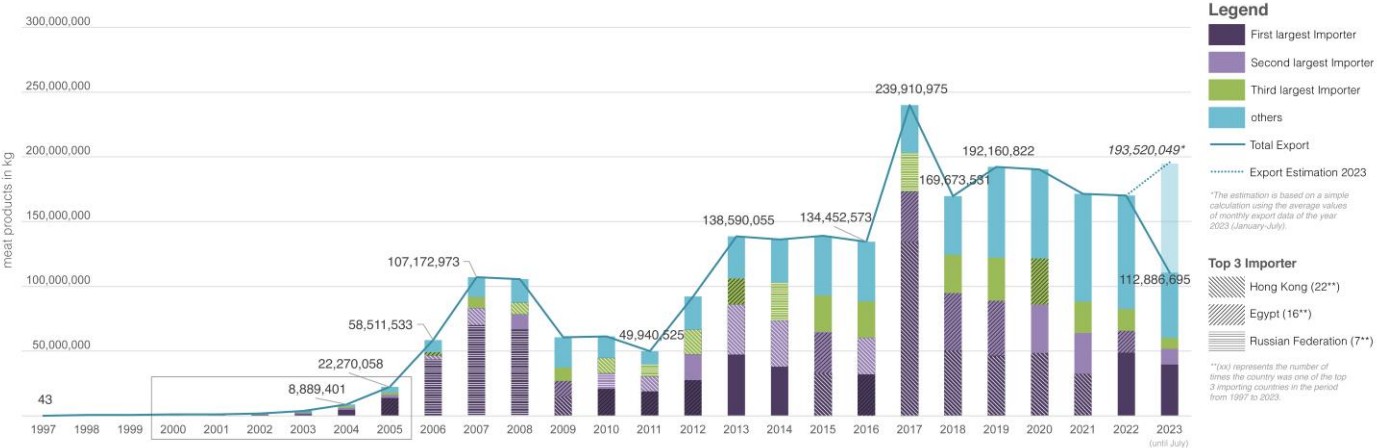

**Figure 4.** Total export of meat products from Rondônia state from 1997 to 2023.

The diversification of the market is striking: the number of countries importing meat products from Rondônia is continuously increasing. In 1997, Rondônia exported to only one country, Bolivia, with a total of 43 kg of meat products; 10 years later, in 2007, there were already 46 countries with a total amount of 107,172,973 kg, and in 2022, there were already 65 countries and 170,083,199 kg of meat products.

In the first 10 years of the records (1997 to 2007), there is an almost exponential increase in exports of meat products. After that, from 2008 onwards, exports stagnated, with export figures declining to the last lowest point in the export history in 2011 at 49,940,526 kg. The reason for this is unknown. From 2012 onwards, a drastic increase can be observed, which, with one exception (2016), continuously rises to the peak of the export numbers in 2017. This can be seen in relation to the deforestation peak in 2016, comparing Figures 3 and 4.

The meat export estimation for 2023 (Figure 4) is based on a calculation using the average values of the first seven months of the year. Based on these data, it appears that 2023 may be the second-highest peak in the history of meat exports from Rondônia.

The total export of meat products for the period of 2000 and 2005, to compare with the deforestation peak of the same period, can be seen in detail in Figures 5 and 6. After the deforestation peak of 2003, a large growth curve of meat product exportation followed, with meat from Rondônia reaching Egypt, Hong Kong, Saudi Arabia and Israel. From 2003 to 2005, the exported meat product volume increased 5.78 times.

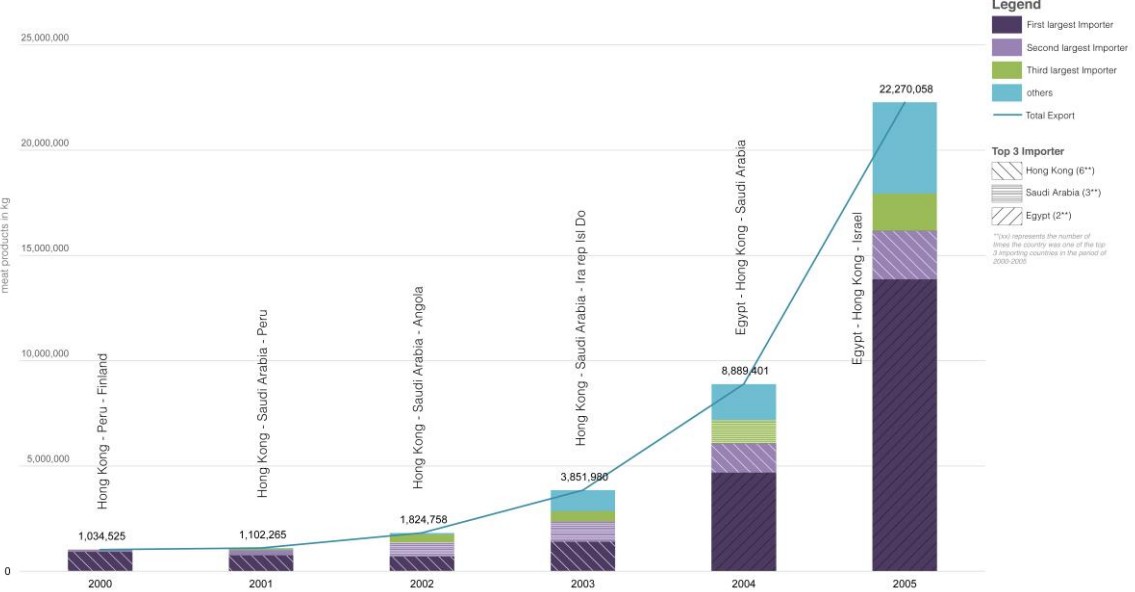

**Figure 5.** Total export of meat products from Rondônia state from 2000 to 2005.

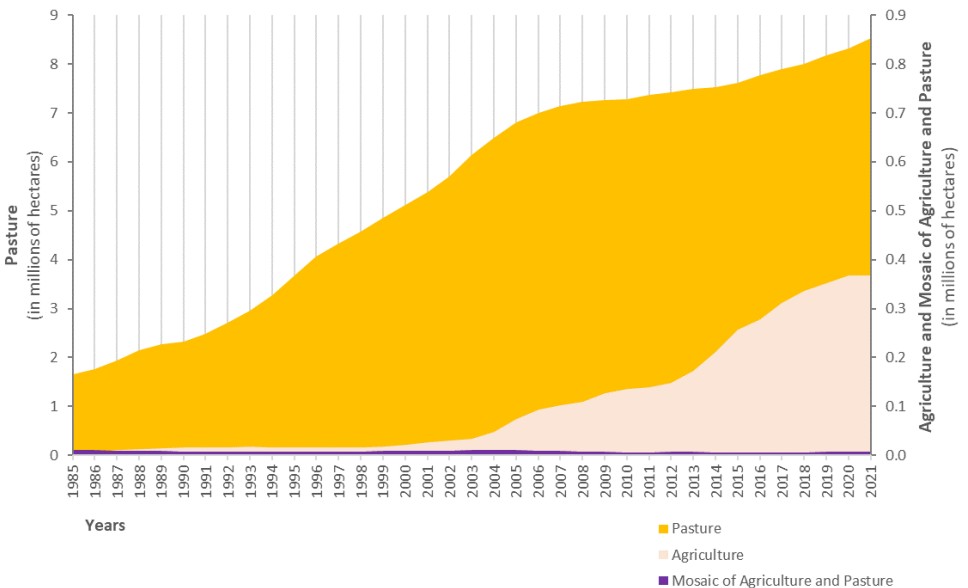

**Figure 6.** Land-use patterns in Rondônia state from 1985 to 2021.

Based on the Mapbiomas imagery, Figure 6 shows the land use pattern in the state of Rondônia from 1985 to 2021 for agricultural activities. In this graph, it is possible to analyze the impressive growth curve of pasture areas up to 2021, and the predominance of this land use in comparison to agriculture.

The data show that the increase in beef production from 1985 to 2023 was accompanied by an increase in the total pasture area associated with beef cattle. This relationship is also visible when analyzing the increase in the number of cattle for the state of Rondônia: the total number of cattle increased from 764,299 head (1985) to 14,349,219 (2019), a massive increase of 1.877% [25,42,43].

Figure 7 shows the relationship between land use change patterns and population data (human and cattle). The figure highlights the reduction in the amount of forest in relation to the increase in pasture and the curves of population and cattle. The proportion of cattle per inhabitant went from 0.85 to almost 8 cattle per inhabitant in 2019. At the same time, satellite imagery shows an increase in the pasture areas in Rondônia from 1,648,428 hectares in 1985 to 8,173,651 hectares in 2019 (+495%). In the same period, urban areas increased from 19,460 hectares to 49,901 hectares (+256%). By visualizing the data compiled in Figure 7, it is evident that the increase in the deforestation of the rainforest (dark green) runs parallel to the increase in pastureland (light green).

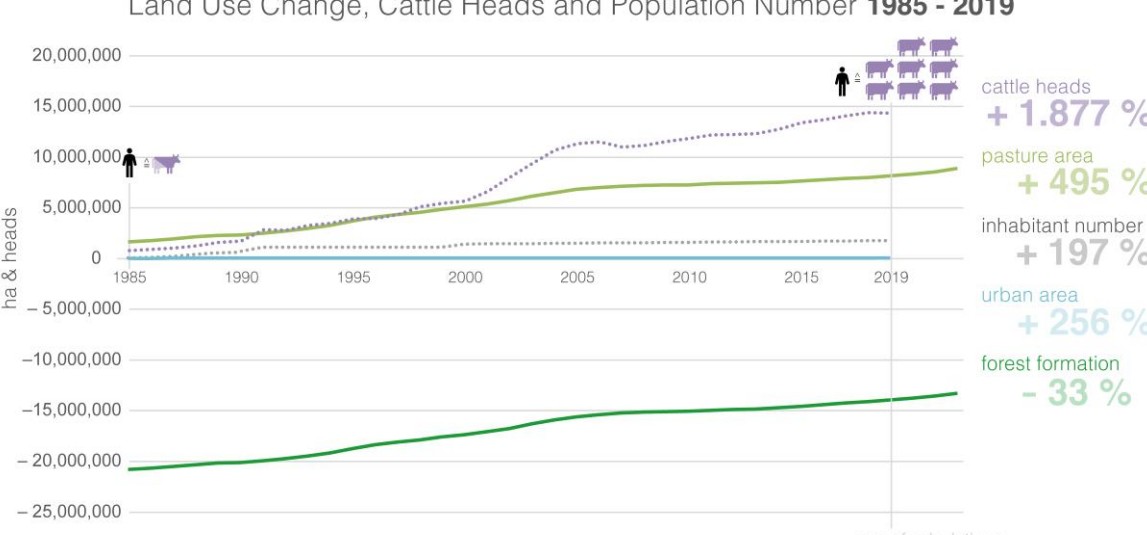

**Figure 7.** Relation between land-use data (forest and pasture areas) and population data (inhabitant and cattle heads) in Rondônia state from 1985 to 2019.

## 4. Discussion

The results indicate that the majority of deforestation in the Amazon region is attributable to human activities, including the expansion of cattle ranging, agriculture and logging. In a smaller measure, mining activities also contribute to deforestation. Furthermore, deforestation is associated with the occurrence of uncontrolled forest fires, which are employed as a means of clearing forests for the illegal expansion of farming activities. In the Brazilian Amazon, 87% of the deforestation can be linked to cattle ranching. This is the case of Rondônia. The findings suggest that deforestation in the Amazon region of Rondônia is directly linked to the expansion of cattle ranching. The map analysis showed that the conversion of forest areas to pasture peaked in 2002–2003. Similarly, the number of cattle has increased steadily since 1985, with an intense period of expansion from 2000 to 2005.

The consumption of animal products involves more than beef and other meat. In addition to these products, it is important to note that rainforest deforestation due to soy production (which is used as animal feed) is increasing significantly. In 1985, soya plantations accounted for 0.0003% of the surface area of the state of Rondônia, and by 2021, this proportion had increased to 1.34%. In the international Amazon, the deforestation associated with soy plantations in Bolivia and in the Brazilian state of Mato Grosso stands

out. It is possible that the soy produced on Amazonian land is also linked to cattle feed products around the world, but this hypothesis needs to be investigated [44].

Most of the meat produced in the state of Rondônia was marketed in the Brazilian national market, as mentioned. An effort was made to understand the fate of the meat produced in Rondônia from 1985 to 2021, but unfortunately, it was not possible to select this period of comparison since data for the national market are not available for all periods.

It was, however, possible to analyze and compare the meat produced in the state of Rondônia for the national and international markets, considering a selection of meat product categories directly related to human consumption, for the period from January 2019 to January 2021 [45]. During this period, a total of approximately 6 billion kilos of meat products related to beef were commercialized in the Brazilian market, of which more than 4 billion kilos were related to products intended for direct human consumption, such as chilled or frozen beef. Of this total, 81% was destined for storage, while 12% went to the most densely populated areas of Brazil (the states of São Paulo and Rio de Janeiro), and 3% was marketed directly in the Amazon region (the states of Rondônia and Amazonas). Unfortunately, it is not possible to estimate the destiny of the meat registered for stock, which accounts for the largest amount of meat sold in the period, at more than 3 billion meat products.

From January 2019 to January 2021, the international market purchased a similar number of meat products as the Brazilian state of São Paulo (376,988,898 kilos of meat products for the international market versus 372,152,598 for the state of São Paulo).

In 2019 and 2020, the biggest buyer of meat products from the state of Rondônia was Hong Kong, which purchased between 43,242,751 and 45,086,878 kilos of beef products. In 2019, the second-largest buyer was Egypt, followed by Chile in third position and the Arab Emirates in fourth. China remained in sixth position that year, but in 2020, it was already the second-biggest buyer, followed by Egypt and Chile.

Figure 8 is a comparison between the national market and the international market. This shows the comparison of meat products from Rondônia state consumed by the international and national markets in 2019 and 2021.

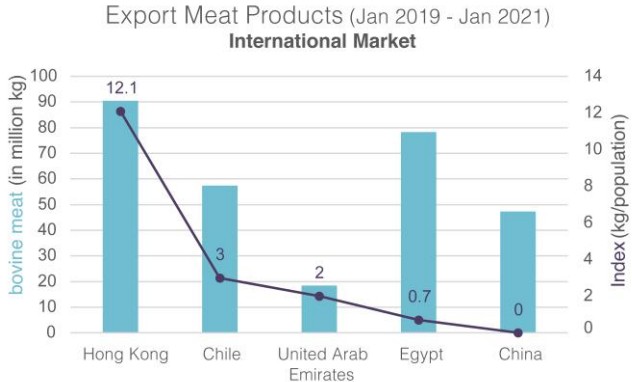 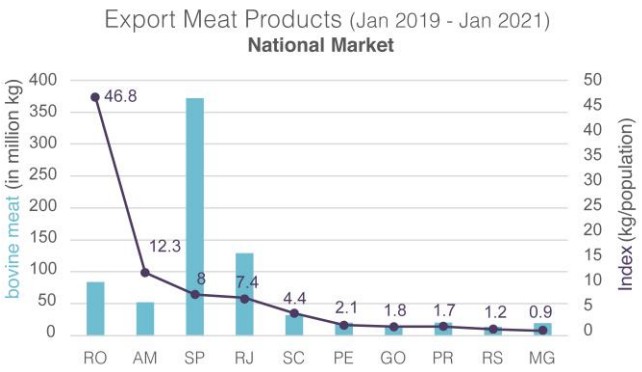

**Figure 8.** Comparison of Rondônia meat products consumed by the international and national markets from January 2019 to January 2021.

To make this comparison possible, it was important to create a consumption ratio (kilos of meat products per population). For this purpose, the average population from 2019 to 2021 was calculated, and an index of kilos of meat per population was created. For the selected period, it is possible to highlight that the highest consumption index came from the state of Rondônia (RO) itself: with an average population of 1,796,321 inhabitants, the state consumed 84,120,334 kilos of meat in the selected period, which represents an index of 46.82.

In second place was the Brazilian state of Amazonas: with an average population of 4,207,435 and a consumption of 51,790,184 kilos of beef products during the period,

the consumption ratio (kg/population) was 12.3. Hong Kong was ranked third, with an average population of 7,497,219 consuming 90,488,578 kilos of meat, giving it a ratio of 12.

In the case of the Brazilian states of São Paulo and Rio de Janeiro, with average populations of 46,285,838 and 17,364,827, the ratios were 8 and 7.4, respectively.

This analysis allows us to understand that the meat produced in the state of Rondônia, when compared to the population data, was mostly consumed by the populations of the Amazon region (in the Brazilian states of Rondônia and Amazon), followed by Hong Kong.

In general, annual meat consumption will vary due to cultural and specific dietary habits, as well as economic and market factors such as product availability and income levels. Research has shown that beef consumption is declining in many countries due to a number of factors, including environmental concerns, changing lifestyles and animal welfare concerns. However, the Food and Agriculture Organization (FAO) of the United Nations reports that global meat consumption of all types is on the rise, increasing by more than 40% between 1990 and 2018 [46].

China, the world's most populous country, is also one of the world's largest consumers of beef. It has also experienced changes in food consumption in recent years, however, with a dietary shift toward more animal products. There are many reasons for this, including the increasing population and urbanization process, income growth and the increasing availability of meat in the markets [47].

A slight decrease in bovine meat production is forecast globally, but an increase in total meat production (bovine, poultry, swine, sheep) is predicted (FAO, 2022). In Brazil, beef production is projected to increase due to increased global import demand and high international cattle prices.

Brazil is classified as an emerging beef market, and it is predicted that Brazil, China, the European Union and the United States will produce approximately 60% of the world's meat by 2029 [36].

Agricultural activities play an important role in the Brazilian economy, and this is no different in the Amazon region. Livestock production in Brazil is divided into two segments: meat and milk. Together, they represent a value of BRL 67 billion (EUR 12 billion) [48]. The exportation of meat represented 3% of Brazilian exports, representing 6% of the gross domestic product (GDP) or 30% of the GDP of agribusiness [49].

Environmental problems are often linked to other problems, such as poverty and the unequal distribution of wealth. Although cattle ranching has been a boon to the economy of the state of Rondônia, this does not mean that the income is evenly distributed. Indeed, according to data, 32% of the population of Rondônia was living in poverty in 2021 [50].

In fact, there is a kind of tolerance of the impacts of the exploitation of natural resources in the Amazon region because it is considered to be good for the economy, but this logic prioritizes only short-term economic growth, partially neglecting long-term consequences for the environment and society [27].

It is also important to point out that the consumption of plant-based foods has been increasing, and the food industry [51] and consumers are looking for strategies to reduce meat consumption. "Flexitarian", vegetarian and vegan diets are associated with health and nutrition benefits [52], ecological concerns [53] and concerns for animal welfare. The conscious and voluntary choice to reduce (flexitarians) or exclude (vegans, vegetarians) some types of animal-based foods from the diet is therefore an increasing global trend.

Additionally, research on veganism and vegetarianism around the world is growing in the scientific community, often linked to health studies [53–58]. Increased awareness of environmental issues, ethics in food production and compassion for animals have made plant-based diets more popular, and vegetarianism and veganism are not just a dietary trend but also a social movement. The prevalence of meatless or plant-based diets varies around the world for many reasons, including cultural and religious beliefs, health habits and ethical concerns.

There is a correlation: foods that are good for our health are also good for our environment in general [53]. Furthermore, the environmental impacts of meat production

and consumption are well documented and include land use, biodiversity, water, air and soil quality.

Technologies, techniques and practices can help reduce emissions: solutions such as artificial meat, methane inhibitors, vertical farming, 3-D printing of food, agroforestry and smart cropping systems with novel perennial plants are being developed and implemented around the world but are not yet widely used [9,59]. In addition, extensive livestock production is often associated with remote environments, where deforestation reflects weak land use policies, and with locations that have cost advantages, such as the availability of large tracts of land, as in the case of the Brazilian Amazon.

There are alternative strategies for reconciling economic growth and natural resource use with greater ecological balance. These include sustainable agricultural development to reduce deforestation and other natural habitat conversions. For livestock, one strategy is to intensify and adopt technological innovations for the beef production system, especially considering the use of land that has already been cleared for pasture [39].

Public policies that focus on the conservation of natural resources and biomes are also key to more sustainable production systems. In Brazil, however, the transition to a sustainable economy is not yet a reality on a large scale. It may therefore be necessary for society to rethink its consumption habits.

As a result of the Brazilian government's commitment to reduce greenhouse gas emissions in the agricultural sector, in 2011 the ABC Plan was created and approved, an initiative to consolidate a low-carbon economy in agriculture, addressing climate change through a series of mitigation and adaptation actions, such as the management of ecosystems and biodiversity, and sustainable production and consumption.

The program developed strategies such as the rehabilitation of degraded pastures and the implementation of integrated systems such as crop-livestock-forest (ICLF), a system approved by law in Brazil in 2013 [39]. These are considered more efficient and sustainable production methods that can reduce pressure on remaining native vegetation areas and reduce GHG emissions.

However, despite the legal framework, technological developments and even public rural credit policies to encourage sustainable practices, the reality in Brazil is still that beef production is based on the historical use of extensive pastures, and newer technologies have not been implemented on a wide scale.

## 5. Conclusions

In this study, we hypothesized global responsibility for the deforestation of the Amazon because Brazil is one of the largest beef exporters in the world. Our results showed that most of the meat produced in Rondônia was used for the domestic market, placing most of the responsibility for deforestation on Brazil. On the other hand, data revealed the growing significance of exports over the years, with key importing countries including Hong Kong, Egypt, Russia and Venezuela. In 1997, Rondônia exported a minimal amount of meat products to only one country, Bolivia. A decade later, exportation exports expanded to 46 countries, totaling 107 million kg of meat products. By 2022, exports rose to 170 million kg, reaching 65 countries.

The findings suggest that deforestation is a global issue, with conscious consumption choices being capable of playing a key role in achieving sustainable population growth and reducing negative environmental impacts.

The research emphasizes the necessity of fostering consensus on sustainable strategies to halt deforestation in the Amazon and the expansion of pastures. One potential avenue for actions is to persuade global consumers of meat products to consider modifying their dietary habits.

Our discussion showed that global food consumption trends point to increased demand for all types of food in the future: Diets that exclude or limit meat consumption are expected to grow in popularity. However, global consumption of all types of meat is also expected to increase. In this sense, our findings support the view that protecting the Ama-

zon Forest is a global responsibility, and that reducing animal-based food consumption is an important strategy in the face of population growth and increasing global consumption.

In this context, it is important to recognize that a global political commitment to the protection of the Amazon is complex, with a history of violent power struggles over the exploitation of the territory and its natural resources, combined with a specific reality of rules and disputes, mostly linked to illegal actions. After a long period when deforestation in the Brazilian Amazon attracted little public attention, meetings have been held to discuss the issue, such as the Amazon Summit 2023. In the European context, measures are also being implemented, such as the European Regulation 2023/1115, published in June 2023, which regulates the export of products linked to deforestation and forest degradation.

However, the international discussion still does not address key aspects of controlling deforestation that require strong political courage, such as the inclusion of environmental criteria in export and import markets by more countries around the word. This is why individual actions, like reducing meat consumption, are key to tackling deforestation.

**Author Contributions:** Conceptualization, V.G.D.; data collection, V.G.D.; satellite imagery analysis, M.C.M.H. and V.G.D.; data collection graphics, C.G.; analysis, V.G.D.; writing—original draft preparation, V.G.D. and M.C.M.H.; writing—review and editing, V.G.D. and M.C.M.H.; supervision, C.R.; funding acquisition, V.G.D. All authors have read and agreed to the published version of the manuscript.

**Funding:** This research had the support of The Alexander von Humboldt Foundation (Experienced Researcher Grant, Georg Forster Fellowship, for V.G.D.).

**Institutional Review Board Statement:** Ethical review and approval were waived for this study since the method was based on systematic observation. Participants were not at risk during the research.

**Informed Consent Statement:** Not applicable.

**Data Availability Statement:** The raw data supporting the conclusions of this article will be made available by the authors on request.

**Acknowledgments:** The authors gratefully acknowledge the support of the Alexander von Humboldt Foundation.

**Conflicts of Interest:** The authors declare no conflicts of interest.

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
