# Peer review of "Amazon Deforestation and Global Meat Consumption Trends: An Assessment of Land Use Change and Market Data from Rondônia That Shows Why We Should Consider Changing Our Diets"

_sustainability, doi:10.3390/su16114526_

Round 1

Reviewer 1 Report

Comments and Suggestions for Authors

ABSTRACT

The authors should

1)    remove lines 10-13 because they are showing the results of their own research as a fact before analyzing the data

2)    rephrase lines 20-22 because the conclusions can not be drawn from their researcher

INTRODUCTION

The introduction shows a clear biased towards meat consumption, authors should restrain themselves of these opinions until they can prove it with their own research.

1)    Lines 31-33 are unsubstantially opinions on the subject; therefore, it should be removed. They already define our food system unsustainable with no proof or references, and so on.

2)    Lines 34-38 should include REFERENCES, if not, please remove

3)    Lines 49-50, some could argue that 12% is not heavily deforestation¡¡¡, ease moderate the biased opinion and show the facts with substantial bibliography

4)    Please give the percentage of this 12% that is linked to cattle ranching¡¡¡

5)    Lines 61-65 are results¡¡¡ they should not be in the introduction

6)    Lines 67-75 are unnecessary. Readers can see this description for themselves

Recommendations

1)    The introduction should include the benefits of the forest and the consequences of their disappearance.

2)    Before imposing the burden directly to the meat consumption, they should discuss the matter in depth and add many other factors involved in the discussion. 

3)    The real objective of the study should be clearly defined in the last paragraph.

2. MATERIALS AND METHODS

The authors do Not describe the calculations (methods) use to account for the losses
There is a wide explanation on the tools (Mapbiomas, Foreing trade statistics, etc…)

However a proper (statistical) analysis of the data is not described

3. RESULTS

            I understand the importance of the work, however the authors fail to expose results in a scientific matter, they mix their results with other type of investigation that is not being treated in this study as:

Lines 154-157---

The Amazon region faces several challenges, such as the increase in deforestation due to the unplanned expansion of agriculture and cattle ranching and illegal activities related to the exploitation and extraction of natural resources, such as mining, timber and its products. To address these challenges, public policies and global action are needed to build a consensus on the sustainable use of native forests.

Or this one

Lines 201-208

Thanks to the existence of protected areas and indigenous territories, deforestation has been more stable since 2018. However, this is not without pressure in these protected areas, which face local conflicts with illegal activities such as slash-and-burn and logging, the first step of deforestation that precedes the introduction of cattle to mark territories. Although recent, these land-use conflicts in the state of Rondônia are not new. The misperception of the value of the rainforest and the creation of a culture of exploitation 207 have been present in Brazil since colonial times, as discussed in the following section

Or this one

Lines 211-214

The first records of occupation of the Amazon territory show the presence of settle- 211 ments approximately 14,000 years ago. They were most likely Asian immigrants, whose 212 arrival on the American continent is explained by a number of hypotheses, such as their 213 arrival via the Bering Strait when the sea level was lower or by the navigations of Chinese 214 admirals on expeditions around the world [21].

THE SAME FROM LINES 201-243

NOT RELATED TO THE SCIENTIFIC TOPIC IN THIS PAPER

PLEASE REVIEW THE REST OF THE TEXT THAT IS NOT A CONSECUENCE OF THE RESEARCH PAPER

Comments on the Quality of English Language

The English used is OK

But as in any other paper, editors must manage this

Author Response

23 March 2024. Letter for Reviewer 1.

Dear reviewer of the Manuscript ID sustainability-2919031,

First of all, we would like to thank you for reviewing our manuscript. We appreciate your comments, which we have taken into account to improve our paper.

Out of respect for your time and effort, we are sending the reviewed article with the changes in red to facilitate a new revision process. We have also added information as suggested by you and the other reviewers.

Considering your comments on our paper, we would like to highlight

1- The title of the manuscript has been rewritten;

2- The abstract has been revised as suggested;

3 - In the introduction, some lines have been removed and others rephrased and corrected, as suggested. More references have also been added to support the analysis;

4 - In the Results section, the suggestion to remove some text was not considered. It is important to note that the exclusion of paragraphs was not recommended by the other two reviewers. We also consider the paragraphs highly relevant, as it provides information to help readers understand our findings. In this sense, we decided to keep all sections of the results in the main text. In addition, the text was revised.

We would again like to thank you for your contribution in this manuscript. We remain at your disposal if further revisions are required.

Cordially,

The authors.

Reviewer 2 Report

Comments and Suggestions for Authors

The manuscript investigated the reason of deforestation in the Amazon, the trends of land use, growth of pasture area and beef production in Brazil especially Rondonia. This manuscript is well designed and has good structure. It is a topic of interest to the researchers in the related areas, but it still needs little revision before acception.

1.     For this manuscript, it mainly discusses deforestation in Brazil, especially Rondonia, so maybe the title should be more focused, like this article: F. Recanati, F. Allievi, G. Scaccabarozzi, T. Espinosa, G. Dotelli, M. Saini, Global Meat Consumption Trends and Local Deforestation in Madre de Dios: Assessing Land Use Changes and other Environmental Impacts, Procedia Engineering, Volume 118, 2015,Pages 630-638, https://doi.org/10.1016/j.proeng.2015.08.496.

2.     Please delete the horizontal line in the background of Fig.1, Fig.3, Fig.4 (not clear enough), Fig.6, Fig.7 and Fig.8, which will make them clearer and more concise.

3.     Fig. 5 shows total export of meat products from Rondonia state from 2000 to 2005. If the author can add new data, like from 2000 to 2021, it will be much better.

4.     Fig.6 represents land-use patterns in Rondonia state from 1985 to 2021, while Fig.7 has relation between land-use data and population data in Rondonia state from 1985 to 2019, the result will be more appealing if Fig.7 also has data from 1985-2021.

5.     As for the conclusion part, I think some content maybe in the discussion will be better. It seems the conclusion part is a little too long.

Comments on the Quality of English Language

The quality of English Language is good.

Author Response

23 March 2024. Reviewer 2.

Dear reviewer of the Manuscript ID sustainability-2919031,

First of all, we would like to thank you for reviewing our manuscript. We appreciate your comments, which we have taken into account to improve our paper.

Out of respect for your time and effort, we are sending the reviewed article with the changes in red to facilitate a new revision process. We have also added information as suggested by you and the other reviewers.

Considering your comments on our paper, we would like to highlight

1- The title of the manuscript has been rewritten, as suggested;

2- The graphics were corrected, as suggested;

3 and 4- Unfortunately, it was not possible to generate further graphics. In the case of Figure 7, this was not possible because the database is not available. Figure 5, on the other hand, should not be shown over a longer period, as it highlights the export of meat products during the period (2000-2005) when deforestation was particularly high, and is therefore representative of the results.

We would again like to thank you for your contribution in this manuscript. We remain at your disposal if further revisions are required.

Cordially,

The authors.

Reviewer 3 Report

Comments and Suggestions for Authors

The presented manuscript deals with deforestation in the Brazilian Amazon in the state of Rondonia. The authors prove the relationship between this process and the increase in pasture areas and beef production. Bibliographic research, Landsat satellite imagery and the MapBiomas methodology for Landsat mosaic and classification were used.

The topic and research plan of the study are interesting. However, in reviewer’s opinion, the manuscript needs some improvements as noted below.

-      Throughout the manuscript, the authors alternately use different units of area: hectares and square kilometers. For easier reading and comparison, I suggest choosing one unit and unify it throughout the manuscript.

-      Please carefully check the presentation of all numbers throughout the manuscript, including graphs, and make appropriate corrections. This concerns the use of uniform decimal and thousand separators in all numbers (in accordance with the guidelines of the MDPI).

-      There are several cases of providing numerical data without citing the source (e.g. lines 42, 44, 260-266). Please complete it.

-      Figures 4-8 are difficult to read - please try to improve the readability of the graphs. Titles within the chart are unnecessary, the description under the figure is sufficient. All charts should have a similar form, e.g. the description of the Y axis in Fig. 4 and 5 should be the same, but they are not: 'meat products in kg', 'Meat products (kg)'. Please use appropriate thousand separators in the numbers of the Y axis. Figure 6 - no description of the Y axis - please complete it, please shorten the description of the X axis markers to only the year.

-      Please check all citations and adapt them to the journal's requirements (e.g. lines 125, 337, 474).

-      Line 374-379 - Hong Kong, Egipt, Chile, China etc.   are not exporters, Brazil (state of Rondonia) is an exporter. Please rephrase the sentences.

-      Lines 377-379 – ‘… it was already the second largest exporter, followed by Egypt and Chile.’ - 'second' or maybe 'third' ?

Author Response

23 March 2024. Reviewer 3.

Dear reviewer of the Manuscript ID sustainability-2919031,

First of all, we would like to thank you for reviewing our manuscript. We appreciate your comments, which we have taken into account to improve our paper.

Out of respect for your time and effort, we are sending the reviewed article with the changes in red to facilitate a new revision process. We have also added information as suggested by you and the other reviewers.

Considering your comments on our paper, we would like to highlight

1- The title of the manuscript has been rewritten, as suggested by some of the reviewers;

2- The text passed English revision, including the decimal and thousand separators and number units;

3- New references were added and adjusted to MDPI guidelines;

4- We decided to keep the graphics in their original format. We feel that they may have been difficult to read due to the low quality of the pdf when generated by the system, but the high-resolution images are clear. We also decided to keep the captions on the images, as we believe that readers often download the images and forget the caption.

We would again like to thank you for your contribution in this manuscript. We remain at your disposal if further revisions are required.

Cordially,

The authors.

Reviewer 4 Report

Comments and Suggestions for Authors

​ 1. What does the research address the main question?

The authors show the growth of cattle ranching as a consequence of the conversion of the Amazon forest into a pasture to meet the growing needs of Brazil and global consumers. The main message is the urgent need for more conscious consumption, production practices, and ethical and sustainable eating habits. It is too general and difficult to understand in practice. Moreover, the abstract and introduction must include the study's aim. In lines 56-59, we probably find any hypothesis, and then we are informed about the manuscript's structure, which better fits the abstract. 

Therefore, it isn't clear if we do not know the aim of the manuscript. 

We find the paper's aim in the Conclusion paragraph: In this study, the authors aim to encourage conscious consumption choices related to eating habits worldwide to help protect the Amazon forest.

2. Do you consider the topic original or relevant in the field? 

The topic is original and very important, and we expect to find answers and solutions to some questions in the manuscript: 

  • why is there increasing demand for beef meat? 
  • is beef important from a quality and health point of view? 
  • why do many customers decide not to consume beef meat because of its high cost and preference to eat cheaper poultry meat?
  •  why does the uncontrolled forestation of the Amazon occur? 

3. What does it add to the subject area compared with other published material?

The Authors point out the need to build consensus on sustainable management to stop or control deforestation of the Amazon and the expansion of pasture lands. It is a fact that both phenomena take place, but how can we counteract them?

4. What specific improvements should the authors consider regarding the methodology? What further controls should be considered?

The methodology is proper and well-presented. 

It includes MapBiomas land analyses, data on beef meat production, and population and livestock data.

5. Discussion

In this paragraph, we would like to know if our results are similar or different from those obtained by other researchers. However, the Authors try to prove that deforestation of the Amazon is responsible for the growing demand for beef meat in Brazil, Hong Kong, Egypt, Russia, and Venezuela. Finally, we are all responsible for it. The second factor is the growing production of soy, which is linked to cattle feeding.

6. Are the conclusions consistent with the evidence and arguments, and do they address the main question?

The conclusions should be consistent with the evidence and arguments. Therefore, you should move some text fragments from this paragraph to another one to explain factors that translate into the growing demand for meat and possibly for change in this area.  

Additionally, thanks to it, you avoid repetitions in text.

You could use SWOT analysis to show the problem's complexity clearly. The conclusion paragraph should include the main ideas of your paper resulting from your experiments ( materials and methods) and your extensive, exciting review. Also, you can support your hypothesis if it is properly constructed (I doubt that my 

low consumption of beef meat, which is very expensive, on the other hand, makes me responsible for the Amazon deforestation. 

7. Are the references appropriate?

Ok

Please check references according to requirements: 3 different presentations?

14. Procedia Engineering. Volume 550 118, 2015, pages 630-638.

20. International Journal of Remote Sensing 32, 4407–4429 (2011).

35. Sustainability 2023, 15, 4670

8. Please include any additional comments on the tables and figures.

They are clear and well-presented.

8. Other comments

The text needs to be rewritten with precise aim and efforts to find connections between factors responsible for the Amazon deforestation, increase in the pasture, global food trends, possible changes of the actual state, and future potential activities to decrease temporary trends.

Author Response

09 May 2024. Reply to Reviewer 4.

Dear reviewer of the Manuscript ID sustainability-2919031,

First of all, we would like to thank you for reviewing our manuscript. We appreciate your comments, which we have taken into account to improve our paper.

Out of respect for your time and effort, we are sending the reviewed article with the changes in red to facilitate a new revision process. We have also added information as suggested by you.

In light of your comments on our paper, we would like to point out that we have paid special attention to the reordering of some paragraphs and the rewriting of parts in order to better connect the text, especially in the discussion and conclusion.

We would again like to thank you for your contribution in this manuscript. We remain at your disposal if further revisions are required.

Cordially,

The authors.

Round 2

Reviewer 1 Report

Comments and Suggestions for Authors

ABSTRACT

The abstract should succinctly encapsulate the key findings of the review, providing a comprehensive overview rather than the cursory and uninformative introduction it currently represents.

INTRODUCTION

The introduction shows a clear biased towards meat consumption, authors should restrain themselves of these opinions until they can prove it with their own research.

1)    Lines 26-27 is an unsustainable statement. The reference cited provides numbers, but it is the authors biased opinion that producing enough meat is difficult.

2)    The introduction begins with global meat consumption data and continues with an incoherent phrase: "Not only animal sources of meat, but also meat quality¡¡¡". Then, the concept of quality and the characteristics used to evaluate it, as well as the factors that affect it, are defined. However, it lacks scientific arguments regarding the possibility of producing enough meat to meet demand, which introduces the topic of cultured meat. Subsequently, the evaluation of serum markers to assess quality and performance in traditional meat production is addressed, suggesting that these methods could also be applied to cultured meat

PLEASE REWRITE THE INTRODUCTION AS IT SHOULD CONTAIN:

a)     An initial paragraph that describes the problem to be addressed by this research.

b)    The context that provides data on how other researchers have currently addressed this problem.

c)     The proposed solution should be outlined.

2. Conventional evaluation of meat quality

The authors are encouraged to expand upon the review provided in this chapter and enhance its completeness by further elaborating on Section 3, titled "Serum Biomarkers for Meat Quality." Currently, Section 2 appears incomplete and overly simplistic. For instance, while myoglobin is discussed as a color indicator and various enzymes are mentioned in relation to tenderness, there is no mention of any serum biomarkers for water-holding capacity (WHC). Additionally, the explanation of beef grading systems seems unnecessary, particularly since it does not delve into serum biomarkers.

Moreover, Section 4, which addresses "Cultured Meat and Limitations," briefly touches on the issue but fails to establish a clear relationship with the rest of the paper or its objective.

Lastly, Section 5 attempts to bridge the topics of the manuscript by exploring the "Potential Application of Serum Markers for Cultured Meat Production." However, it primarily reiterates information from previous sections without introducing any novel insights. The conclusions also fall short of providing new information that should have emerged from the review. While the authors propose the addition of serum markers to the culture media as a cutting-edge and secure method for enhancing the production and quality of cultured meat, this assertion lacks substantive evidence to support its validity. Thus, further research and evidence are needed to substantiate this claim.

Comments on the Quality of English Language

no comments

Author Response

09 May 2024. Reply to Reviewer 1.

Dear reviewer of the Manuscript ID sustainability-2919031,

We believe that there may be an error in the comments uploaded to the system. It is clear that these comments refer to a different paper that discusses meat quality and cultured meat production. We wrote to the Sustainability Editorial Board on May 2 about this concern, but we did not receive any feedback from them. Therefore, it was only appropriate to proceed with the review of the paper based on Reviewer 4's comments.

We would like to point out that we have paid particular attention to reordering some paragraphs and rewriting parts to better connect the text, especially in the discussion and conclusion, as suggested by reviewer 4. The changes are marked in red in the text.

We would again like to thank you for your contribution in this manuscript. We remain at your disposal if further revisions are required.

Cordially,

The authors.

Reviewer 4 Report

Comments and Suggestions for Authors

​ 1. What does the research address the central question?

The paper explores land use changes in Rondônia and the growth of pasture areas and beef production. This sentence is not the aim of the paper. It is a method, and the following sentences are ok in the abstract.

 The aim of the paper still needs to be included. I want you to write the sentence: the paper's aim is…..

 In the Introduction paragraph, the sentence: "The article presents findings about the growth of cattle ranching in the Amazon region, which is a consequence of the conversion of the rainforest into pasture. A case study is presented for the Brazilian state of Rondônia. In my opinion, it is not the aim of the paper. 

Line 60: I do not understand the sentence: …" that meat from this Amazon region is linked to deforestation and consumed worldwide."

Line 64: sentence:  "The research emphasizes the necessity of fostering consensus on sustainable strategies to halt deforestation in the Amazon and the expansion of pastures. One potential avenue for action is to persuade global consumers of meat products to consider modifying their dietary habits," would be moved to conclusions.

Line 71: the sentence "The findings suggest that deforestation is a global issue, with conscious consumption choices capable of playing a key role in achieving sustainable population growth and reducing negative environmental impacts" as above should not be in the Introduction paragraph.

2. Do you consider the topic original or relevant in the field? 

The topic is original and very important.

3. Materials and Methods are adequately written with clear explanations.

4. Results. Subparagraphs concerning the history of Rondônia, cattle ranching, and the international market can be acceptable, as it is your work to give basic information to present your experimental results.

The following explanations are very well presented and improve the quality of the manuscript.

5. Conclusions

They are clear and well-written.

6. Are the references appropriate?

They were extended and connected with some more appropriate sentences in the text.

7. Please include any additional comments on the tables and figures.

They are clear and well-presented.

Comments on the Quality of English Language
  1. I am not qualified in English, but if I checked the text by Grammarly Premium, I found many options for English improvements. 

Example: Abstract

Abstract: This paper explores land use changes in Rondônia and the growth of pasture areas and beef production. Brazil is one of the largest beef producers in the world, with products destined for local and global markets. Based on bibliographic research, the paper analyzes maps of land use change between 1985 and 2021, using Landsat satellite imagery and the MapBiomas methodology for Landsat mosaic and classification. Research shows that beef from Rondônia is primarily purchased and consumed in Brazil but is also bought and sold internationally. Landsat imagery analysis shows the predominance of forest conversion to pasture in Rondônia. The results show that deforestation in the Amazon is directly linked to the growth of cattle ranching. Land use change from native forest to pasture for beef cattle production is a reality in the Brazilian Amazon, especially in Rondônia. It suggests the urgent need for more conscious consumption and production practices as well as ethical and sustainable eating habits.

Author Response

14 May 2024. 

Dear reviewer of the Manuscript ID sustainability-2919031,

We would like to thank you for reviewing our manuscript. We appreciate your comments, which we have taken into account to improve our paper. We are sending the reviewed article with the changes in red.

We would again like to thank you for your contribution in this manuscript. We remain at your disposal if further revisions are required.

Cordially,

The authors.

Round 3

Reviewer 1 Report

Comments and Suggestions for Authors

The manuscript improved greatly

Author Response

14 May 2024. Reviewer 1.

Dear reviewer of the Manuscript ID sustainability-2919031,

We would again like to thank you for your contribution in this manuscript. We remain at your disposal if further revisions are required.

Cordially,

The authors.
